# The Prognostic Significance of the Pan-Immune-Inflammation Value in Patients with Heart Failure with Reduced Ejection Fraction

**DOI:** 10.3390/diagnostics15131617

**Published:** 2025-06-25

**Authors:** Emir Dervis, Idris Yakut, Duygu Inan

**Affiliations:** 1Department of Cardiology, Medipol University, 34815 Istanbul, Turkey; 2Department of Cardiology, Sincan Training and Research Hospital, 06949 Ankara, Turkey; idrislive@windowslive.com; 3Department of Cardiology, Basaksehir Cam and Sakura City Hospital, 34480 Istanbul, Turkey; duyguinan@gmail.com

**Keywords:** heart failure, systolic, inflammation, prognosis, biomarkers, mortality

## Abstract

**Objective:** We aimed to investigate the association between the pan-immune-inflammation value (PIV) and mortality in patients with heart failure with a reduced ejection fraction (HFrEF), along with clinical and biochemical parameters. **Methods**: In this retrospective cohort study, 419 patients diagnosed with HFrEF between January 2014 and December 2023 were analyzed. Data on demographic features, comorbidities, cardiac parameters [New York Heart Association (NYHA) classification, left ventricular ejection fraction (LVEF), ventricular dimensions], medication use, and laboratory findings (PIV, N-terminal pro-B-type natriuretic peptide [NT-proBNP], electrolytes, and complete blood count) were collected from institutional and national records. **Results**: Mortality occurred in 22.91% of patients. PIV > 696 was significantly associated with mortality (sensitivity: 37.5%, specificity: 78.64%, *p* = 0.006), but it was not an independent predictor in multivariate analysis. Instead, low body mass index (BMI), increased end-systolic diameter, reduced LVEF, advanced NYHA class (III/IV), elevated NT-proBNP, hyponatremia, and lymphopenia were identified as independent predictors (all *p* < 0.001). **Conclusions**: Although PIV was associated with mortality in patients with HFrEF, it did not independently predict outcomes beyond established risk factors. These results suggest that while inflammation may contribute to HFrEF pathophysiology, traditional clinical and biochemical markers remain more reliable for prognostication.

## 1. Introduction

Heart failure (HF) describes a condition in which the heart cannot sufficiently carry out its pumping function, resulting in blood flow and hemodynamics that do not meet the metabolic demand [1,2]. This condition can cause symptoms such as shortness of breath, fatigue, reduced exercise tolerance, and fluid accumulation in the lungs, abdomen, and lower extremities. It is associated with high morbidity and mortality rates, significantly impacting the quality of life and functional capacity of affected individuals [3].

Heart failure is commonly categorized into three subtypes: heart failure with a reduced ejection fraction (HFrEF; LVEF < 41%), mildly reduced ejection fraction (LVEF 41–49%), and preserved ejection fraction (HFpEF; LVEF ≥ 50%) [4]. Among these, HFrEF is characterized by adverse cardiac remodeling, neurohormonal activation, and systemic inflammation, which contribute to disease progression.

Inflammation plays a significant role in the pathophysiology of HFrEF, as continuous cardiac stress leads to the increased release of pro-inflammatory cytokines, oxidative stress, and endothelial dysfunction [2,5]. Various immune-inflammatory biomarkers derived from peripheral blood counts, such as lymphocyte, neutrophil, monocyte, and platelet counts (PLR) [6,7], the neutrophil-to-lymphocyte ratio (NLR) [8], and the systemic immune-inflammation index (SII) [9], have been associated with poor prognosis in HF. The SII is calculated using the formula SII = (Neutrophils × Platelets)/Lymphocytes, and it reflects the balance between inflammatory and immune response [9]. However, these markers assess only partial cell counts and have certain limitations, resulting in low predictive value [10].

The pan-immune-inflammation value (PIV) is a more comprehensive indicator of systemic inflammation calculated using the following formula: PIV = (Neutrophils × Platelets × Monocytes)/Lymphocytes [10,11]. PIV reflects the inflammatory burden more effectively relative to other easily accessible markers such as the neutrophil-to-lymphocyte ratio, platelet-to-lymphocyte ratio, and SII [11]. PIV has been shown to have prognostic significance in various diseases, including colorectal cancer, melanoma, breast cancer, non-small-cell lung cancer, antineutrophil cytoplasmic antibody-associated vasculitis, hidradenitis suppurativa, and peritoneal dialysis [10,12,13,14,15,16]. However, data on its association with HFrEF prognosis are limited.

This study aims to investigate the relationship between PIV and mortality in patients with HFrEF, along with other clinical parameters (including age, sex, body mass index [BMI], smoking status, comorbidities, cardiac characteristics [ischemic cardiomyopathy, cardiac device implantation, echocardiographic parameters, New York Heart Association (NYHA) class], and biochemical parameters [N-terminal pro b-type natriuretic peptide (NT-proBNP), electrolytes, renal function tests, lipid profile, complete blood count indices]).

## 2. Material and Methods

### 2.1. Study Design, Setting and Participants

This study was performed in the Cardiology outpatient clinics of Ankara City Hospital, Ankara and Istanbul Medipol University, Istanbul, Turkey, by retrospectively examining patients diagnosed with HFrEF from January 2014 to December 2023. The diagnosis, follow-up and treatment procedures of HFrEF were based on current European Society of Cardiology Guidelines definitions. Patients younger than 18 years of age, those diagnosed with septic shock, active inflammatory disease or infection at the time of diagnosis (or within 3 months prior), those with a known malignancy history at the time of diagnosis, pregnant women, those with missing data regarding the variables included in the study and patients lost to follow-up were excluded. Patients were classified and compared based on mortality (survivor vs. deceased), and analyses were performed to identify factors associated with this primary outcome. This study was approved by the Non-Interventional Clinical Research Ethics Committee of Istanbul Medipol University (Decision date: 13 June 2024, decision no: 578). All procedures performed were in accordance with the ethical standards of the Helsinki declaration and its later amendments.

### 2.2. Collection of Cardiac and Other Data

The data used for comparisons were obtained/calculated from the time at diagnosis. These included age and sex, body mass index (BMI), smoking status (non-smoker, ex-smoker or active smoker), comorbidities, the rate of ischemic cardiomyopathy, use of cardiac devices [implantable cardioverter defibrillator (ICD) or cardiac resynchronization therapy (CRT)], echocardiographic measurements (end-diastolic diameter, end-systolic diameter, and LVEF), NYHA classification, medications, follow-up durations, mortality data, and laboratory findings [N-terminal pro b-type natriuretic peptide (NT-proBNP), sodium, glucose, urea, serum creatinine, serum albumin, low-density lipoprotein cholesterol (LDL-C), triglyceride, high-density lipoprotein cholesterol (HDL-C), total cholesterol, and hemoglobin levels, absolute neutrophil, lymphocyte, monocyte, and platelet counts], which were retrospectively obtained from the hospital computerized database and the national database records (e-nabız). Patients were stratified based on device therapy: those with an ICD—including both primary and secondary prevention devices—were classified as the device therapy group, regardless of device activation during follow-up, while patients without ICD implantation comprised the control group.

Left ventricular function was assessed by calculating the ejection fraction using echocardiographic 2D measurements according to the modified Simpson’s rule with biplane planimetry. LVEF was defined as the percentage ratio of the difference between end-systolic and end-diastolic volumes to the end-diastolic volume [17].

The laboratory analyses were performed by the Biochemistry Department of Istanbul Medipol University, using calibrated devices and original kits, and according to the manufacturer’s recommendations. PIV was calculated using the following formula: Neutrophil count (10^3^/µL) × Platelet count (10^3^/µL) × Monocyte count (10^3^/µL)]/Lymphocyte count (10^3^/µL) [17]. For NT-proBNP, we calculated log NT-proBNP (Logarithmic NT-proBNP) values with the following formula: log NT-proBNP = ln(NT-proBNP). We calculated eGFR using the Chronic Kidney Disease Epidemiology Collaboration (CKD-EPI) creatinine equation (2021) [18].

### 2.3. Statistical Analysis

Statistical analyses of collected data were carried out with SPSS, version 25.0 (Windows operating system) (IBM Corp., Armonk, NY, USA). For significance, a <0.05 value was accepted as the threshold for significant findings. The normality of continuous variable distributions was checked with histograms and Q-Q plots. Descriptive statistics for continuous data included mean ± standard deviation (SD) and median (25–75th percentile), the former being for normally distributed variables and the latter being for non-normally distributed variables. Frequency (percentage) was used for categorical variables. Between-group comparisons were based on Student’s t or the Mann–Whitney U test, again depending on normality. Categorical variables were analyzed with the chi-square tests or Fisher’s exact tests. Multivariable logistic regression analysis (forward conditional selection method) was performed to determine the significant variables independently associated with mortality. Statistically significant variables according to univariate analysis results were included into the multivariable model.

## 3. Results

A total of 419 patients were included in the study. The mean age of all patients was 55.09 ± 13.54 years, and 78.76% (*n* = 330) were male. Mortality was observed in 22.91% (n = 96). There were no significant differences in age (*p* = 0.403) and sex (*p* = 1.000) between deceased and surviving patients. The mean BMI in the mortality group was significantly lower (*p* < 0.001). The median follow-up duration was 29 (17–71) months. In the mortality group, the frequencies of hypertension (*p* = 0.003) and atrial fibrillation (*p* = 0.001) were significantly lower, as were the frequencies of patients with NYHA class I or class II (*p* < 0.001), and those not requiring device therapy (*p* = 0.040). Conversely, patients who reached mortality had higher frequencies of NYHA class III and class IV (*p* < 0.001), as did those requiring device therapy (*p* = 0.040), as well as higher median BNP levels (*p* < 0.001), mean logNT-proBNP levels (*p* < 0.001), and median PIVs (*p* = 0.006) (Table 1).

**Table 1 diagnostics-15-01617-t001:** Summary of patient and disease-related characteristics and laboratory measurements with regard to mortality.

		Mortality		
	Total (*n* = 419)	Yes (*n* = 96)	No (*n* = 323)	*p*
Age, years	55.09 ± 13.54	56.10 ± 12.48	54.79 ± 13.84	0.403 ^†^
Sex				
Female	89 (21.24%)	20 (20.83%)	69 (21.36%)	1.000 ^#^
Male	330 (78.76%)	76 (79.17%)	254 (78.64%)	
Body mass index, kg/m^2^	27.05 ± 3.83	25.48 ± 3.42	27.51 ± 3.82	<0.001 ^†^
Smoking status				
Non-smoker	279 (66.59%)	69 (71.88%)	210 (65.02%)	0.445 ^#^
Ex-smoker	49 (11.69%)	10 (10.42%)	39 (12.07%)	
Active smoker	91 (21.72%)	17 (17.71%)	74 (22.91%)	
Diabetes mellitus	147 (35.08%)	34 (35.42%)	113 (34.98%)	0.938 ^#^
Hypertension	173 (41.29%)	27 (28.13%)	146 (45.20%)	0.003 ^#^
Atrial fibrillation	271 (64.68%)	48 (50.00%)	223 (69.04%)	0.001 ^#^
Ischemic cardiomyopathy	206 (49.16%)	46 (47.92%)	160 (49.54%)	0.781 ^#^
Cardiac device	314 (74.94%)	75 (78.13%)	239 (73.99%)	0.493 ^#^
ICD	305 (72.79%)	74 (77.08%)	231 (71.52%)	0.466 ^#^
CRT	9 (2.15%)	1 (1.04%)	8 (2.48%)	
End-diastolic diameter, mm	65.13 ± 10.21	66.84 ± 11.74	64.62 ± 9.67	0.061 ^†^
End-systolic diameter, mm	54.87 ± 10.02	57.27 ± 10.31	54.15 ± 9.83	0.007 ^†^
LVEF	24.30 ± 6.60	21.42 ± 6.32	25.15 ± 6.45	<0.001 ^†^
NYHA classification				
Class I	36 (8.59%)	2 (2.08%)	34 (10.53%)	<0.001 ^#^
Class II	250 (59.67%)	44 (45.83%)	206 (63.78%)	
Class III	120 (28.64%)	41 (42.71%)	79 (24.46%)	
Class IV	13 (3.10%)	9 (9.38%)	4 (1.24%)	
Device therapy				
None	135 (61.09%)	21 (46.67%)	114 (64.77%)	0.040 ^#^
Therapy	86 (38.91%)	24 (53.33%)	62 (35.23%)	
Medications				
MRA	206 (49.16%)	45 (46.88%)	161 (49.85%)	0.609 ^#^
ACE inhibitors/ARB	355 (84.73%)	75 (78.13%)	280 (86.69%)	0.059 ^#^
Beta blockers	414 (98.81%)	95 (98.96%)	319 (98.76%)	1.000 ^§^
Digoxin	95 (22.67%)	25 (26.04%)	70 (21.67%)	0.369 ^#^
Follow-up time, months	29 (17–71)	20 (11–64.5)	38 (18–74)	0.001 ^‡^
NT-proBNP	1126 (435–2310)	1920.5 (1130–3373)	967 (362–2037)	<0.001 ^‡^
logNT-proBNP	2.97 ± 0.56	3.24 ± 0.46	2.89 ± 0.57	<0.001 ^†^
Sodium, mEq/L	136.97 ± 4.50	134.69 ± 5.64	137.65 ± 3.85	<0.001 ^†^
Glucose, mg/dL	105 (90–131)	102 (92–128.5)	106 (89–131)	0.543 ^‡^
Urea, mg/dL	34 (24–47)	36 (24–54.5)	34 (24–46)	0.375 ^‡^
Creatinine, mg/dL	1.03 (0.86–1.23)	1.04 (0.85–1.26)	1.02 (0.86–1.23)	0.544 ^‡^
eGFR, mL/min/1.73 m^2^	79.85 ± 25.33	77.74 ± 26.76	80.48 ± 24.89	0.353 ^†^
Albumin, g/dL	3.81 ± 0.43	3.69 ± 0.44	3.85 ± 0.42	0.002 ^†^
LDL-C, mg/dL	100.72 ± 35.20	92.91 ± 32.35	103.04 ± 35.72	0.013 ^†^
Triglyceride, mg/dL	116 (89–158)	102 (76–138.5)	126 (92–178)	<0.001 ^‡^
HDL-C, mg/dL	37 (30–43)	36 (26–41.5)	38 (31–44)	0.054 ^‡^
Total cholesterol, mg/dL	167.18 ± 45.06	153.61 ± 43.50	171.21 ± 44.79	0.001 ^†^
Hemoglobin, g/dL	13.38 ± 2.05	12.97 ± 1.94	13.51 ± 2.07	0.024 ^†^
Neutrophil (×10^3^)	5.13 (4.00–6.30)	5.30 (3.80–6.46)	5.10 (4.06–6.27)	0.600 ^‡^
Lymphocyte (×10^3^)	1.95 ± 0.80	1.62 ± 0.73	2.05 ± 0.80	<0.001 ^†^
Monocyte (×10^3^)	0.65 ± 0.25	0.65 ± 0.24	0.65 ± 0.26	0.868 ^†^
Platelet (×10^3^)	237.30 ± 74.16	233.49 ± 75.28	238.43 ± 73.91	0.567 ^†^
PIV (×10^6^)	345.09 (227.50–697.92)	441.97 (251.99–911.50)	334.80 (219.37–632.46)	0.006 ^‡^

Descriptive statistics are presented using mean ± standard deviation for normally distributed continuous variables, median (25–75th percentile) for non-normally distributed continuous variables and frequency (percentage) for categorical variables. ^†^, Student’s *t* test; ^‡^, Mann–Whitney U test; ^#^ chi-square test; ^§^, Fisher’s exact test. Abbreviations: ACE: angiotensin-converting enzyme; ARB: angiotensin receptor blockers; BNP: brain natriuretic peptide; CRT: cardiac resynchronization therapy; eGFR: estimated glomerular filtration rate; HDL-C: high-density lipoprotein cholesterol; ICD: implantable cardioverter defibrillator; LDL-C: low-density lipoprotein cholesterol; LVEF: left ventricular ejection fraction; MRA: mineralocorticoid receptor antagonist; NYHA: New York Heart Association; PIV: pan-immune-inflammation value. The PIV, with a cut-off value of >696, was able to significantly distinguish mortality in patients with HFrEF [sensitivity = 37.50%, specificity = 78.64%, AUC (95% CI) = 0.592 (0.527–0.658), *p* = 0.006] (Table 2), (Figure 1).

**Table 2 diagnostics-15-01617-t002:** Performance of the pan-immune-inflammation value to predict mortality and ROC curve analysis.

Cut-off	>696
Sensitivity	37.50%
Specificity	78.64%
Accuracy	69.21%
PPV	34.29%
NPV	80.89%
AUC (95% CI)	0.592 (0.527–0.658)
*p*	0.006

Abbreviations: AUC: area under ROC curve; CI: confidence interval; NPV: negative predictive value; PPV: positive predictive value; ROC: receiver operating characteristic.

Multivariable logistic regression analysis results had revealed that low BMI (OR: 0.856, 95% CI: 0.788–0.930, *p* < 0.001), high end-systolic diameter (OR: 1.029, 95% CI: 1.002–1.057, *p* = 0.035), low LVEF (OR: 0.925, 95% CI: 0.886–0.966, *p* < 0.001), NYHA class III and IV (OR: 1.948, 95% CI: 1.121–3.385, *p* = 0.018), high log NT-proBNP (OR: 2.198, 95% CI: 1.179–4.097, *p* = 0.013), low sodium (OR: 0.896, 95% CI: 0.848–0.947, *p* < 0.001) and low lymphocyte count (OR: 0.629, 95% CI: 0.432–0.915, *p* = 0.015) were independently associated with mortality. Other variables included in the analysis, hypertension (*p* = 0.092), atrial fibrillation (*p* = 0.244), albumin (*p* = 0.065), LDL-C (*p* = 0.393), triglyceride (*p* = 0.148), total cholesterol (*p* = 0.539), hemoglobin (*p* = 0.799) and PIV (*p* = 0.873), were found to be non-significant (Table 3).

## 4. Discussion

The results of the current study demonstrated that a high PIV (>696) demonstrated a significant association with mortality in ROC analysis; however, it was not an independent predictor in multivariate analysis, suggesting its role may be limited as a screening marker. Low BMI, high end-systolic diameter, low LVEF, NYHA classes III and IV, high log NT-proBNP, low sodium levels, and low lymphocyte count were identified as risk factors for mortality.

Heart failure was identified as an emerging epidemic around 25 years ago. Currently, with the population both growing and aging, the number of individuals affected by heart failure continues to rise [19]. Over the past 30 years, improvements in treatment methods and their implementation have increased survival and reduced hospitalization rates for patients with HFrEF [17]. Despite this progress, HFrEF remains a significant public health issue and continues to impose a substantial economic burden [4]. Accurately predicting the prognosis of this condition plays a crucial role in determining the appropriate type and timing of treatment. The well-known biomarker BNP is extensively used to accurately assess prognosis in heart failure patients, but it has notable limitations in clinical practice. BNP levels often fall short of expected values due to their short half-life. Consequently, researchers are increasingly focused on identifying new biomarkers that are more cost-effective and easier to access [17].

In this context, inflammatory markers have gained increasing attention as potential prognostic indicators in heart failure [17,20]. Studies have highlighted that inflammation plays a central role in the development and progression of HFrEF, and various blood-derived markers reflecting systemic inflammation have been evaluated for their predictive value. For instance, composite indices such as the NLR, PLR, SII, and PIV have been proposed as accessible and inexpensive prognostic tools.

Emerging evidence suggests that composite scores derived from blood counts, which can be associated with inflammation, may serve as prognostic biomarkers in heart failure [20]. However, many of these markers have limited clinical applicability, and precision is a matter of constant debate [17]. HFrEF is strongly associated with immunological and inflammatory responses in the recent literature [17,20]. This study aimed to investigate the relationship between PIV, a more sensitive inflammatory biomarker, and mortality in HFrEF patients, shedding light on the association between HFrEF prognosis and inflammation. Although univariate analysis indicated that PIVs were significantly higher in deceased HFrEF patients compared to survivors, multivariate analysis did not identify PIV as an independent predictor of mortality. While ROC analysis suggested that high PIV (>696) could significantly predict mortality in HFrEF patients, its utility in predicting mortality in HFrEF appears limited in this cohort.

The relationship between heart failure and inflammation has been investigated in numerous previous studies. A growing body of evidence suggests that composite inflammatory indices derived from routine blood tests may offer superior prognostic value compared to single inflammatory markers in HF. These studies have demonstrated that inflammatory biomarkers such as C-reactive protein [21], tumor necrosis factor-alpha (TNF-α) [22], interleukin (IL)-6 [23], lymphocyte, neutrophil, monocyte, and platelet counts [6,7], NLR [8], and SII [9] are associated with short- and long-term prognosis in heart failure. PIV has been associated with the prognosis of coronary artery disease [11], hypertension [24], and abdominal aortic calcification [25], as well as the incidence of the coronary slow flow phenomenon [26] and major cardiovascular and cerebrovascular events [27]. Inan et al. investigated the prognostic value of the PIV in patients with decompensated acute heart failure. They found that patients with high PIVs had higher in-hospital mortality rates compared to those with low PIVs. PIV has been a stronger predictor of prognosis in acute heart failure patients compared to other well-known inflammatory markers [20]. In another study, Murat et al. demonstrated that a higher PIV was associated with 5-year all-cause mortality, as well as in-hospital mortality in patients with HFrEF hospitalized for acute decompensated heart failure [17]. Notably, while the PIV shows strong prognostic performance in acute HF settings [17,20], our study’s null finding in chronic HFrEF suggests that the following: (1) the inflammatory burden may be more predictive during acute decompensations than stable phases, and (2) the predictive power of inflammatory markers may depend on the clinical context and HF phenotype.

Lymphocytes, neutrophils, monocytes, and platelets are the main cells involved in the response to infection and inflammation [6,28]. PIV is a variable that integrates these four cell counts, and therefore may be a more accurate representation of inflammation [6,11,29]. Our study found that while the PIV was not associated with the prognosis of HFrEF, lymphocyte count was, warranting further studies to assess whether lymphocytes exert a greater impact on heart failure. Lymphocytes are important components of the immune system; lower levels can indicate a weakened immune system and increased vulnerability to infections, and may contribute to the progression of many chronic diseases, including heart failure [30]. Additionally, a low lymphocyte count may signal the presence of other comorbidities, which can further increase mortality risk. Patients with a low lymphocyte count typically have poorer nutritional status, lower physical capacity, and overall weaker health profiles [31]. These factors can increase the risk of mortality in patients with heart failure. In our study, the use of all-cause mortality as the primary outcome may have contributed to these results. These findings highlight the need for special attention and closer monitoring of HFrEF patients with a low lymphocyte count in clinical practice. Furthermore, our results underline the potential for lymphocyte count alone, rather than composite indices, to act as a simpler yet powerful marker of immune competence and prognosis in HFrEF. However, further research is needed to explore the relationship between the PIV, lymphocyte count, and the prognosis of HFrEF.

In this study, we investigated factors associated with all-cause mortality in patients with HFrEF. Low BMI, high end-systolic diameter, low LVEF, NYHA class III and IV, high NT-proBNP, and low sodium levels were identified as independent risk factors. Excess body weight increases the workload on the heart and can exacerbate heart failure symptoms. Additionally, high BMI is associated with comorbid conditions such as hypertension, diabetes, and coronary artery disease, which can further contribute to increased mortality in heart failure. Low BMI often indicates malnutrition, which reduces the body’s capacity for recovery and fighting infections. This can lead to worse outcomes in heart failure patients. Heart failure can also enhance the inflammatory response, leading to muscle wasting and weight loss. Patients with low BMI may be more vulnerable to the effects of these catabolic processes [32]. An increase in the end-systolic diameter of the left ventricle indicates ventricular dilation and dysfunction, leading to reduced pumping capacity and poorer clinical outcomes, worsening heart failure symptoms and increasing mortality [33]. NYHA classes III and IV indicate severe symptoms and substantial limitations in daily activities among patients with heart failure, representing an anticipated result [34]. The detection of higher levels of NT-proBNP as an independent factor is another expected result, because this peptide released in response to ventricular stress in heart failure patients. Elevated NT-proBNP levels indicate increased cardiac stress and worsening heart failure, associated with poor prognosis and higher mortality risk [35]. Low sodium levels typically reflect advanced heart failure and poor renal function. Additionally, low sodium levels are associated with fluid and electrolyte imbalances, which may increase mortality risk. While these are well-established risk factors, our study also aimed to contribute to the growing body of literature focusing on inflammatory biomarkers. Given the central role of inflammation in heart failure pathophysiology, incorporating such markers into prognostic models may enhance risk stratification beyond conventional clinical parameters. Previous studies have demonstrated that several clinical parameters—including high BMI [36], ventricular dysfunction and increased chamber volumes [37], reduced LVEF [38,39], higher NYHA class [36,40,41,42], elevated BNP levels [38,42], and hyponatremia [42]—are significant risk factors for mortality. Besides the well-established risk factors reported in previous studies and supported by our findings, a multitude of other risk factors have been linked to poor prognosis in heart failure, including demographics, genetic and environmental factors, medications, markers of catabolic processes, blood count parameters, comorbidities impacting the heart, cardiac echocardiography and electrocardiography findings, and various novel markers measured in the circulation [39,40,41,42,43,44,45,46]. However, many studies have overlooked the possibility that these risk factors may differ between HFrEF and HFpEF. Various studies have supported that HFrEF and HFpEF may have different pathophysiological mechanisms and, consequently, different risk factors [38,45].

### Study Limitations

This study is one of the few investigating the relationship between PIV and mortality in patients with HFrEF. Although no clear positive results were found in favor of PIV, the findings suggest that the results reported in previous studies require further investigation, and the inflammatory characteristics of the pathophysiology of HFrEF need more research. However, this study has several notable limitations. The retrospective design of this study has led to inherent limitations such as selection bias and missing data, which, along with factors like the relatively small sample size and the single-center approach, restrict the generalizability of the results. Mortality events were recorded until August 2024, and thus, patients included in the later stages of the study underwent a shorter follow-up, which could have biased the results. Most importantly, due to the retrospective nature of the data collection process, information regarding treatments that could impact inflammatory burden were ignored, which could have had considerable impacts on PIVs and other inflammation-related parameters. Inflammatory markers including PIV were measured only at the time of initial HFrEF diagnosis, potentially not capturing dynamic changes during disease progression or treatment. The absence of a healthy control group and an HFpEF patient group prevented comparisons between these groups and HFrEF patients. The primary outcome being all-cause mortality is another significant limitation; however, data availability limited the evaluation of mortality directly attributable to heart failure.

## 5. Conclusions

The present study reports an all-cause mortality rate of 22.91% during a ten-year period in patients with HFrEF. Despite a high PIV (>696) being a significant predictor of mortality in HFrEF patients with low sensitivity, it was not an independent risk factor for mortality, suggesting its potential utility in identifying patients at higher risk. Independent predictors of mortality were BMI, end-systolic diameter, LVEF, NYHA class, log NT-proBNP, and sodium levels. The relationship between PIV and prognosis in HFrEF patients and the inflammatory pathophysiology of HFrEF appear to warrant further investigation. This could contribute to the identification of high-risk patients and the development of new treatments, thereby improving long-term outcomes for patients with HFrEF.

## Figures and Tables

**Figure 1 diagnostics-15-01617-f001:**
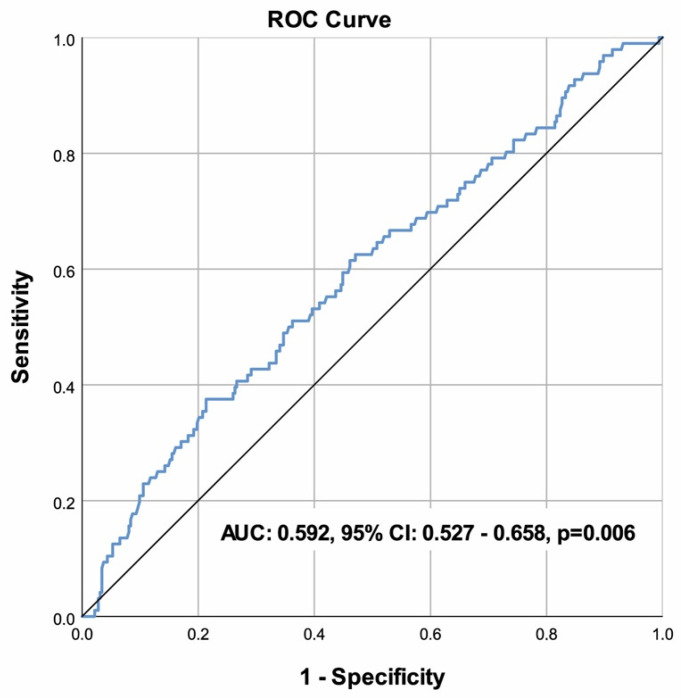
ROC curve of the PIV to predict mortality.

**Table 3 diagnostics-15-01617-t003:** Significant variables independently associated with mortality, multivariable logistic regression analysis.

	β Coefficient	Standard Error	*p*	Exp(β)	95% CI for Exp(β)	
Body mass index, kg/m^2^	−0.156	0.042	<0.001	0.856	0.788	0.930
End-systolic diameter, mm	0.029	0.014	0.035	1.029	1.002	1.057
LVEF	−0.078	0.022	<0.001	0.925	0.886	0.966
NYHA classification, class III and IV	0.667	0.282	0.018	1.948	1.121	3.385
log NT-proBNP	0.788	0.318	0.013	2.198	1.179	4.097
Sodium, mEq/L	−0.109	0.028	<0.001	0.896	0.848	0.947
Lymphocyte (×10^3^)	−0.464	0.191	0.015	0.629	0.432	0.915
Constant	16.199	4.261	<0.001			

Nagelkerke R^2^ = 0.343. Abbreviations: BNP: brain natriuretic peptide; CI: confidence interval; LVEF: left ventricular ejection fraction; NYHA: New York Heart Association.

## Data Availability

The datasets used and/or analyzed during the current study are available from the corresponding author upon reasonable request.

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
