# Peer review of "The Prognostic Significance of the Pan-Immune-Inflammation Value in Patients with Heart Failure with Reduced Ejection Fraction"

_diagnostics, 2025, doi:10.3390/diagnostics15131617_

Round 1
Reviewer 1 Report
Comments and Suggestions for Authors
Prognosis estimation for clinical outcomes such as
morbidity, mortality, and hospitalization plays an important
role in helping patients, their families, and clinicians
decide on the appropriate type and timing of treatment
(especially decisions regarding the rapid transition to
further treatments). In recent decades, although numerous
prognostic markers have been identified to predict death
and/or HF hospitalization for different patient populations
with HF, some of them are useful in predicting death, but
they fall short in predicting hospitalizations. Moreover, their
clinical applicability is limited, and precise risk stratification
in HF remains difficult.
Although the specific pathogenesis of HF remains unclear,
abnormal immune activation and chronic inflammation play
an important role, and it has been proven that there is a
close relationship between inflammation and cardiovascular
diseases.Studies have shown that inflammation plays
an important role in the initiation and progression of
atherosclerosis and is closely associated with the pathogenesis
of HF and cardiac remodeling.Growing evidence also
proposes that immunological and inflammatory responses may
play a pathogenic role in the development of chronic HF(Murat 2024).PIV is a potent predictor of prognosis with better performance than other well-known inflammatory markers for patients with AHF. (Inan 2023).
Author Response
Dear Editor and Reviewer,
Thank you very much for your thorough review of our manuscript and for your contributions. We appreciate your positive assessment of our study and have carefully considered your suggestions to further enhance the manuscript. Below, we outline our responses to each point:
Comments 1:
Prognosis estimation for clinical outcomes such as morbidity, mortality, and hospitalization plays an important role in helping patients, their families, and clinicians decide on the appropriate type and timing of treatment (especially decisions regarding the rapid transition to further treatments). In recent decades, although numerous prognostic markers have been identified to predict death and/or HF hospitalization for different patient populations with HF, some of them are useful in predicting death, but they fall short in predicting hospitalizations. Moreover, their clinical applicability is limited, and precise risk stratification in HF remains difficult. Although the specific pathogenesis of HF remains unclear, abnormal immune activation and chronic inflammation play an important role, and it has been proven that there is a close relationship between inflammation and cardiovascular diseases. Studies have shown that inflammation plays an important role in the initiation and progression of atherosclerosis and is closely associated with the pathogenesis of HF and cardiac remodeling. Growing evidence also proposes that immunological and inflammatory responses may play a pathogenic role in the development of chronic HF(Murat 2024).PIV is a potent predictor of prognosis with better performance than other well-known inflammatory markers for patients with AHF. (Inan 2023).
Response 1:
We would like to highlight that we have already addressed the role of PIV and other inflammatory biomarkers in heart failure prognosis.
Thank you once again for your constructive feedback and we hope the revised version meets the expectations of the journal.
Reviewer 2 Report
Comments and Suggestions for Authors
Abstract part is confusing, I suggest rewriting it since it is non-informative. In the journals Instructions for the authors, you can find all information needed.
From the Introduction part I suggest removing the sentence on lines 36-38, or rephrase.
HF with mid-range LVEF is now designated as HF with mildly reduced ejection fraction (please see updated ESC guidelines).
I suggest to define precisely PIV and SII in the Introduction part.
In the study aims please add all clinical and biochemical parameters that were used beside PIV to investigate prognosis in HFrEF patients.
Please rephrase lines 90 and 91 (Materials and Methods part). Is there a therapy group? Were patients divided into groups according to implanted ICD pacemakers? Why ATP is mentioned here – it is a mode of pacing so it is rather confusing. This part could be confusing for the reader, so please rephrase.
The Discussion part should be improved. Low BMI, high end-systolic diameter, low LVEF, NYHA classes III and IV, high NT-proBNP, low sodium levels are well known markers of poor prognosis in HF, so the focus should be on the studies that examined inflammatory markers in HF prognosis.
In the Study limitations please add that inflammatory markers were measured at the time of first diagnosis of HFrEF.
Comments on the Quality of English LanguageCould be improved.
Author Response
Dear Editor and Reviewer,
Thank you very much for your thorough review of our manuscript and for your contributions. We appreciate your assessment of our study and have carefully considered your suggestions to further enhance the manuscript. Below, we outline our responses to each point:
Comments 1: Abstract part is confusing, I suggest rewriting it since it is non-informative. In the journals Instructions for the authors, you can find all information needed.
Response 1: We have revised the abstract in line with your suggestion.
Comments 2:From the Introduction part I suggest removing the sentence on lines 36-38, or rephrase.
Response 2: The sentence on lines 36-38 has been removed.
Comments 3:HF with mid-range LVEF is now designated as HF with mildly reduced ejection fraction (please see updated ESC guidelines).
Response 3: As suggested, we have revised the text to reflect the current terminology
I suggest to define precisely PIV and SII in the Introduction part.
Response: We have revised the Introduction section to include precise definitions of both the SII and PIV along with their calculation formulas.
Comments 4: In the study aims please add all clinical and biochemical parameters that were used beside PIV to investigate prognosis in HFrEF patients.
Response 4: We have revised the text to explicitly include all clinical and biochemical parameters that were analyzed in relation to prognosis in HFrEF patients.
Comments 5:Please rephrase lines 90 and 91 (Materials and Methods part). Is there a therapy group? Were patients divided into groups according to implanted ICD pacemakers? Why ATP is mentioned here – it is a mode of pacing so it is rather confusing. This part could be confusing for the reader, so please rephrase.
Response 5: We have revised the text to clarify that patients were grouped based on ICD implantation status.
Comments 6: The Discussion part should be improved. Low BMI, high end-systolic diameter, low LVEF, NYHA classes III and IV, high NT-proBNP, low sodium levels are well known markers of poor prognosis in HF, so the focus should be on the studies that examined inflammatory markers in HF prognosis.
Response 6: In response to the suggestion that more emphasis should be placed on inflammatory markers rather than well-established prognostic factors such as low BMI, high end-systolic diameter, low LVEF, NYHA class, NT-proBNP, and sodium levels, we have revised the discussion accordingly.
Comments 7:In the Study limitations please add that inflammatory markers were measured at the time of first diagnosis of HFrEF.
Response 7: We have added clarification that inflammatory markers (including PIV) were measured at baseline diagnosis only, which may limit insight into their longitudinal relationship with outcomes.
Thank you once again for your constructive feedback and we hope the revised version meets the expectations of the journal.
Round 2
Reviewer 2 Report
Comments and Suggestions for Authors
Authors adequately responded to all raised concerns and significantly improved their paper. I suggest publishing.
Please explain the abbreviation SII, that is: "Systemic Immune-inflammation Index" when mentioned for the first time in the text (line 49).
Author Response
Dear Reviewer,
Thank you very much for your valuable time and insightful comments regarding our manuscript.
we would like to kindly note that the abbreviation SII (Systemic Immune-Inflammation Index) was defined upon its first mention in line 48 of the manuscript.
Kind Regards